# How Dopamine Influences Survival and Cellular Immune Response of *Rhipicephalus microplus* Inoculated with *Metarhizium anisopliae*

**DOI:** 10.3390/jof7110950

**Published:** 2021-11-10

**Authors:** Thaís Almeida Corrêa, Jéssica Fiorotti, Emily Mesquita, Laura Nóbrega Meirelles, Mariana Guedes Camargo, Caio Junior Balduino Coutinho-Rodrigues, Allan Felipe Marciano, Vânia Rita Elias Pinheiro Bittencourt, Patrícia Silva Golo

**Affiliations:** 1Programa de Pós-Graduação em Ciências Veterinárias, Instituto de Veterinária, Universidade Federal Rural do Rio de Janeiro, Rio de Janeiro 23897-000, Brazil; thaisalmeida_tac@yahoo.com.br (T.A.C.); jeskvni@gmail.com (J.F.); emily_mesquita@hotmail.com (E.M.); laura-meirelles@hotmail.com (L.N.M.); marigcamargo@gmail.com (M.G.C.); vaniabit@gmail.com (V.R.E.P.B.); 2Campus Nova Iguaçu, Nova Iguaçu, Escola de Medicina Veterinária, Universidade Estácio de Sá, Rio de Janeiro 26220-099, Brazil; caio-jr@hotmail.com; 3Lallemand Plant Care, Patos de Minas 38706-420, Brazil; allanfmarc@gmail.com; 4Departamento de Parasitologia Animal, Instituto de Veterinária, Universidade Federal Rural do Rio de Janeiro, Rio de Janeiro 23897-000, Brazil

**Keywords:** biological control, entomopathogenic fungi, hemocytes, ticks, phagocytic activity, phenoloxidase

## Abstract

Dopamine (DA) is a biogenic monoamine reported to modulate insect hemocytes. Although the immune functions of DA are known in insects, there is a lack of knowledge of DA’s role in the immune system of ticks. The use of *Metarhizium anisopliae* has been considered for tick control, driving studies on the immune response of these arthropods challenged with fungi. The present study evaluated the effect of DA on the cellular immune response and survival of *Rhipicephalus microplus* inoculated with *M. anisopliae* blastospores. Exogenous DA increased both ticks’ survival 72 h after *M. anisopliae* inoculation and the number of circulating hemocytes compared to the control group, 24 h after the treatment. The phagocytic index of tick hemocytes challenged with *M. anisopliae* did not change upon injection of exogenous DA. Phenoloxidase activity in the hemolymph of ticks injected with DA and the fungus or exclusively with DA was higher than in untreated ticks or ticks inoculated with the fungus alone, 72 h after treatment. DA was detected in the hemocytes of fungus-treated and untreated ticks. Unveiling the cellular immune response in ticks challenged with entomopathogenic fungi is important to improve strategies for the biological control of these ectoparasites.

## 1. Introduction

Ticks are bloodsucking ectoparasites that can transmit pathogenic agents to humans and animals. The southern cattle tick *Rhipicephalus microplus* (Acari: Ixodidae) is considered a major concern for livestock producers in tropical areas. This tick causes economical losses estimated at 3.24 billion dollars per year only in Brazil [1]. These losses are due to cattle mortality, the lower market value of leather, and reduced weight gain, milk, and meat production [2,3]. The inappropriate use of synthetic acaricides increases concerns about human and environmental health and advances the selection of resistant tick populations [4,5]. The use of entomopathogenic fungi to control ticks, specifically *Metarhizium* (Hypocreales: Clavicipitaceae) and *Beauveria* (Hypocreales: Cordycipitaceae), has high potential [6,7,8,9,10,11]. These fungi infect the arthropod target upon contact, using their propagules to actively penetrate the host cuticle [12].

The immune response of arthropods is activated when they are challenged with pathogens. This response can be classified as humoral and cellular immune responses. The humoral response involves hemaglutination processes and the production of antimicrobial peptides while the cellular immune response is related to hemocytes and non-specific reactions, such as phagocytosis, nodulation, and encapsulation [13,14,15]. Five distinct hemocytes were described in *R. microplus* ticks: prohemocytes, plasmatocytes, granulocytes, spherulocytes, and oenocytoids [16]. Phagocytic activity and fungal cytotoxicity were reported in hemocytes from *R. microplus* treated with *Metarhizium* [16]. Nevertheless, the cellular immune response of ticks infected with entomopathogenic fungi has not been completely disclosed.

There are several strategies used by the immune response of arthropods to prevent cell death [15]. One is the connection between the immune and nervous systems [17,18]. The nervous system integrates sensory information and sends it to other systems, including the immune system [17]. Linking the nervous and the immune systems are biogenic monoamines, such as dopamine (DA). DA signaling can contribute to the early activation of insects hemocytes [19]. The DA synthesized and released by the hemocytes in insects can act in an autocrine way, supporting or stimulating the phagocytic activity, and increasing the total number of hemocytes [18,19]. DA is also related to ticks’ salivary secretion, known to be a potent activator of salivation, acting through two different receptors [20,21].

The phenoloxidase (PO) activity in hemocytes is another important immune response of invertebrates [22,23]. In insects, the activation of PO is essential for wound healing and the recognition of foreign material during encapsulation and melanization [24,25]. Melanin originates from phenylalanine, which is first hydroxylated to phenylalanine hydroxylase by tyrosine. Tyrosine is then hydroxylated by PO to produce DA. DA oxidizes dopaquinone, which is immediately converted to dopachrome [26]. Dopachrome is structurally rearranged, producing 5–6 dihydroxy, which is oxidized to form indolequinones that are polymerized to eumelanin [26]. In addition to its fundamental role in the immediate immune response of insects, melanin participates in cuticle sclerotization and pigmentation [25,26]. PO activity has been studied in *Ornithodoros moubata* (Acari: Argasidae) [27] and *Rhipicephalus sanguineus* (Acari: Ixodidae) [28]. The activity of PO in the hemolymph of ixodid ticks requires investigation, especially regarding the effect of DA on the activity of PO in ticks infected with entomopathogenic fungi. Accordingly, the present study aimed to evaluate the effect of DA on the survival of ticks treated with *M. anisopliae* and the role of DA in the cellular immune response of *R. microplus* adults inoculated with this fungus.

## 2. Materials and Methods

### 2.1. Ticks

Fully engorged *R. microplus* females were collected from the floor of cattle pens holding artificially infested calves at the Wilhelm Otto Neitz Parasitological Research Station at Federal Rural University of Rio de Janeiro (UFRRJ), Brazil (CEUA/Veterinary Institute, UFRRJ, Seropédica, Brazil—protocol nº 9714220419). After collection, ticks were washed in tap water and immersed into 0.05% sodium hypochlorite solution for three minutes, then dried and identified.

### 2.2. Entomopathogenic Fungus

*Metarhizium anisopliae* LCM S04 [29] was used in the present study. The cultures were grown on oat medium under controlled conditions [25 ± 1 °C; relative humidity (RH) ≥ 80] for 14 days. For the production of blastospores, 3 mL of conidial aqueous suspensions at 10^8^ mL^−1^ were inoculated into 42 mL of Adamek’s medium and incubated at 25 °C for 72 h. The propagules were washed using 0.01% Tween 80 (Isofar, Rio de Janeiro, Brazil) (*v*/*v*) sterile distilled water solution according to the adapted protocol described by Bitencourt et al. [30]. Blastospores were quantified and adjusted to 10^6^ blastospores mL^−1^. Prior to bioassays, an aliquot of 10 μL of blastospores was transferred to potato dextrose agar (PDA) and incubated at 25 ± 1 °C and RH ≥ 80% to evaluate the fungal viability. The germination of propagules was determined 24 h after incubation. As the present study accessed Brazilian genetic heritage, the research was registered at the National System for the Management of Genetic Heritage and Associated Traditional Knowledge (Sisgen) under the code AA47CB6.

### 2.3. Effect of Dopamine on the Survival of Ticks Treated with M. anisopliae

Fully engorged females were divided into five groups with 15 ticks each: C (control/untreated group), P (ticks injected with phosphate buffer solution), D [ticks injected with 3 µL of 1025 ng µL^−1^ DA (Sigma-Aldrich, St. Louis, MO, USA)], M (ticks inoculated with *M. anisopliae* (3 µL; 10^6^ blastospores mL^−1^)), and DM [ticks injected with DA (3 µL; 1025 ng µL^−1^) and after 20 min, *M. anisopliae* (3 µL; 10^6^ blastospores mL^−1^)]. After injection, using an insulin needle and syringe (1 mL; 13 × 0.45 mm) (Descarpack, Ilhota, Brazil) between the scutum and capitulum [14], tick females were incubated at 27 ± 1 °C and RH ≥ 80%. Female mortality was analyzed 24, 48, and 72 h post-injection (hpi). Experiments were performed twice.

### 2.4. Hemocytes Quantification after M. anisopliae Infection

Fully engorged *R. microplus* females were divided into homogeneously weighted groups of 60 ticks each and treated. The groups were C, P, D, M, and DM as previously described. Ticks were injected as described in Section 2.3. The collection of hemolymph was performed 24 hpi after the treatment, according to De Paulo et al. [31]. Hemocytes were quantified in a Neubauer chamber (Kasvi, São José dos Pinhais, Brazil) using an optical microscope (E200; Nikon, Tokyo, Japan). Experiments were carried out in triplicate and repeated twice.

### 2.5. In Vitro Phagocytic Assay

The phagocytic index (ratio of phagocytic and non-phagocytic hemocytes) was calculated based on tick hemocytes harvested from females submitted to different treatments and then exposed in vitro to *M. anisopliae* or Zymosan A (*Saccharomyces cerevisiae*) (Sigma-Aldrich, St. Louis, MO, USA)). Homogeneously weighted groups with 30 fully engorged ticks each were injected with DA (3 µL; 1025 ng µL^−1^) (D) or phosphate buffer (3 µL) (P). Ticks in the control group (C) had no treatment/injection. Injection was performed according to Fiorotti et al. [14]. The hemolymph was collected [13] in 500 μL L-15 Leibovitz medium (Sigma-Aldrich, St. Louis, MO, USA) 24 hpi. The phagocytosis assay followed the protocol established by Kuklinski et al. [32]. The hemocytes were quantified in a Neubauer chamber, adjusted to 2 × 10^4^ cells on a circular coverslip, and placed in a 24-well culture plate (Kasvi, São José dos Pinhais, Brazil) that was incubated at 32 °C for 10 min. *M. anisopliae* blastospores (20 μL; 10^6^ blastospores mL^−1^) or Zymosan A (20 μL; 10^6^ yeasts mL^−1^), used as control, were added to the plate, and the final volume (250 μL) was completed with L-15 Leibovitz medium. The plate (ticks hemocytes plus fungi) was incubated at 32 °C for two hours. The cells were fixed with Methanol (Emsure-Merk, Darmstadt, Germany) for three minutes and stained with Giemsa (Sigma-Aldrich, St. Louis, MO, USA) [33]. To quantify the phagocytic activity for each treatment, at least 6 slides (100 hemocytes per slide) were examined at 1000 × using an optical microscope (E200; Nikon, Tokyo, Japan). The entire experiment was carried out in triplicate and performed three times.

### 2.6. Phenoloxidase Activity in the Hemolymph of R. microplus Inoculated with M. anisopliae

*R. microplus* engorged females were divided into five groups of 27 homogenously weighted ticks each: C, P, D, M, and DM. Females were inoculated between the scutum and capitulum, following the dose described in Section 2.3, using a stereoscopic microscope and microinjector (Drummond, Broomall, PA, USA). Aliquots of 2 μL of hemolymph were collected [13] from each tick 24, 48, and 72 hpi after fungus inoculation and incubated for 10 min with 28 μL of 0.01 M cacodylate buffer (Sigma-Aldrich, St. Louis, MO, USA) and 0.5 mM CaCl_2_ pH 7.0, in flat-bottomed 96-well plates (Kasvi, São José dos Pinhais, Brazil). Ten microliters of a saturated solution of L-DOPA (Sigma-Aldrich, St. Louis, MO, USA) at 4 mg/mL were added to the mixture for 20 min at room temperature. Absorbance was measured on an ELISA plate reader (Thermo Fisher, Waltham, MA, USA) at 490 nm. The cacodylate buffer previously described was used as the blank solution, according to Feitosa et al. [28]. Experiments were carried out in triplicate and repeated twice.

### 2.7. Detection of Dopamine in Hemocytes of R. microplus Treated with M. anisopliae

Four experimental groups with 25 ticks each were evaluated: C, D, M, and DM. The females were inoculated between the scutum and capitulum using a microinjector (Drummond, Broomall, PA, USA). The hemolymph was collected [13] in 500 μL L-15 Leibovitz medium (Sigma-Aldrich, St. Louis, MO, USA), 24 hpi. The hemocytes were quantified in a Neubauer chamber, adjusted to 2 × 10^4^ cells on a circular coverslip, and placed in a 24-well culture plate (Kasvi, São José dos Pinhais, Brazil). The hemocytes were fixed with 4% paraformaldehyde (Sigma-Aldrich, St. Louis, MO, USA) for 30 min and washed in PBS three times. Hemocytes were incubated with primary anti-dopamine antibody (ab6427; Abcam, Cambridge, UK) for 72 h and with the secondary antibody [Goat anti-rabbit Alexa Fluor 594 (red); Invitrogen, Waltham, MA, USA)] for one hour at room temperature. The hemocytes’ nuclei were stained with DAPI (blue) at room temperature, and the hemocytes were observed under a fluorescence microscope (BX 51; Olympus, Tokyo, Japan) according to the protocol adapted described by Wu et al. [19]. DA deposits were quantified using ImageJ 1.52a software (National Institute of Health, Bethesda, MD, USA), and the results are presented as the percentage of marked area. The experiment was performed twice.

### 2.8. Statistical Analysis

Except for survival analysis, data were checked for normality using a Shapiro–Wilk test. PO activity data were square-root transformed before the analysis to better meet the assumptions of normality and homogeneity of variance. Quantification of hemocytes (Section 2.4) and quantification of DA (Section 2.7) were analyzed by one-way ANOVA followed by the Tukey’s test (*p* < 0.05), phagocytic index was analyzed by Kruskal–Wallis test followed by the Dunn’s test (*p* < 0.05), and PO activity was analyzed by two-way ANOVA followed by the Tukey’s test (*p* < 0.05). The tick survival was analyzed using the Log-rank test. Statistical analysis was performed using GraphPad Prism version 8.4.2 for Windows (GraphPad Software, San Diego, CA, USA).

## 3. Results

### 3.1. Effect of Dopamine on the Survival of Ticks Treated with M. anisopliae

Untreated ticks (C) and the ticks treated with PBS (P) or DA (D) had 100% survival. There was a statistical difference between the control groups (C, P, and D) and DM (*p* = 0.0012) or M (*p* < 0.0001). The group inoculated exclusively with the fungus (M) reached 17 ± 7% tick survival 72 hpi, while ticks injected with both DA and *M. anisopliae* (DM) exhibited 67 ± 13% survival 72 hpi (Figure 1A). Ticks injected with DA and treated with *M. anisopliae* had higher survival than ticks inoculated exclusively with *M. anisopliae* (*p* < 0.0001).

### 3.2. Hemocytes Quantification after M. anisopliae Infection

Untreated ticks (C) and ticks injected with PBS (P) exhibited 1.6 ± 0.1 × 10^6^ and 1.6 ± 0.2 × 10^6^ hemocytes mL^−1^ in their hemolymph, respectively, at 24 hpi. Inoculation with *M. anisopliae* blastospores (M) did not reduce the number of hemocytes in comparison to untreated ticks (C) (Figure 1B). Nevertheless, the injected of DA, followed or not by the entomopathogenic fungus (i.e., D or DM), increased the number of hemocytes in comparison to the control group (*p* = 0.003 and *p* < 0.0001, respectively) (Figure 1B).

### 3.3. In Vitro Phagocytic Assay

The phagocytic index was determined two hours after incubation as a ratio of phagocytic and non-phagocytic hemocytes with *M. anisopliae* blastospores or *S. cerevisiae* (Zymozan A). Phagocytic indexes ranged from 24 ± 0.8% (P *M. anisopliae*) to 29 ± 1.4% (D *M. anisopliae*) two hours after incubation and were statistically similar for all the groups that were tested (F_5,12_ = 5; *p* = 0.101) (Figure 1C).

### 3.4. Phenoloxidase Activity in the Hemolymph of R. microplus Inoculated with M. anisopliae

The PO activity in the hemolymph of *R. microplus* fully engorged females was evaluated 24, 48, and 72 hpi of exogenous DA and *M. anisopliae.* The statistical analyses compared different treatments at the same time (24, 48, or 72 h). All groups (C, P, D, M, and DM) exhibited similar PO activity at 24 hpi (*p* > 0.05) (i.e., C = 0.052 ± 0.007 U; *p* = 0.045 ± 0.010 U; D = 0.056 ± 0.007 U; M = 0.034 ± 0.002; DM = 0.033 ± 0.003 U). At 48 hpi, the highest PO activity (0.057 ± 0.010 U) was observed in the hemolymph of ticks injected with PBS (P). Their activity was statistically similar to C (0.043 ± 0.006 U), D (0.044 ± 0.007 U), and DM (0.049 ± 0.004 U) but different to ticks inoculated with *M. anisopliae* (0.029 ± 0.002 U) (*p* = 0.037). At 72 hpi, the PO activity of ticks injected with exogenous DA and *M. anisopliae* (DM) (0.041 ± 0.019 U) was higher than the activity exhibited by the control group (C) (0.009 ± 0.002 U) (*p* = 0.001) and the group inoculated exclusively with *M. anisopliae *(M) (0.012 ± 0.002 U) (*p* = 0.001). The same was observed for the group injected exclusively with exogenous DA (D) (0.032 ± 0.006 U) in comparison to C (*p* = 0.002) and M (*p* = 0.035) (Figure 2).

### 3.5. Detection of Dopamine in Hemocytes of R. microplus Treated with M. anisopliae

*Rhipicephalus microplus* hemocytes were immunolabeled using an antibody against dopamine, and a secondary antibody (Alexa Fluor 594; red) and their nuclei were counterstained with DAPI (blue). As a result, Figure 3A–D show DA granules labeled (red) in the cytosol of hemocytes, enabling an analysis of the presence of dopamine in the cells. Hemocytes from all groups (including untreated ticks) exhibited DA labels. Hemocytes from ticks injected exclusively with exogenous DA (D) (4.6 ± 0.3% intensity) exhibited higher labeling than the other treatments (C: 2.2 ± 0.6% intensity; M: 1.3 ± 0.3% intensity; DM: 2.8 ± 0.3% intensity) (*p* < 0.05) (Figure 3E).

## 4. Discussion

Entomopathogenic fungi were the first agents applied in the microbial control of insects [34]. Usually, entomopathogenic fungi act by infecting their targets through specialized spores (conidia) that attach, germinate, and penetrate the cuticle of the arthropod host. Once infected, the arthropod can challenge the pathogen with its humoral and cellular immunity. Biogenic monoamines, such as DA, are considered the main link between insects’ nervous and immune systems. Despite this, as far as we know, there are no studies on the action of dopamine in the immune system of ticks challenged with entomopathogens.

In the present study, the effect of dopamine on the survival of *R. microplus* inoculated with *M. anisopliae* was accessed. The fungal isolate used here is considered virulent for *R. microplus* ticks [29]. Inoculation (not immersion) of the entomopathogenic fungal blastospores was used to achieve the level of fungus infection required. Blastospores are yeast-like vegetative cells produced in vitro and not identical but analogs to the ones produced in the host’s hemocoel during fungal infection [35]. Based on our results, exogenous DA increased *R. microplus* survival to the fungus almost four times at 72 hpi (Figure 1A). Similar results were reported in a study with *Chilo suppressalis* (Lepidoptera: Crambidae), where larvae were inoculated with DA and immersed in *Beauveria bassiana* suspension [19]. Their results showed that DA increased larval survival 1.2 times 5 to 10 days after the fungus treatment. Wu et al. [19] suggested that, in insects, DA may act in an autocrine, or perhaps paracrine, mechanism to potentiate cellular defense reactions to infection.

Hemocytes are circulating cells and their number in the hemolymph may vary depending on the sex, age, and stage of the arthropod’s development. Additionally, they may be attached to organs or more available in specific situations, such as activation of the immune system or molting [36]. The number of circulating hemocytes in the hemolymph of ticks can change due to fungal infection [31]. In the present study, *M. anisopliae* inoculation with blastospores did not reduce the hemocyte concentration contrary to what was observed by De Paulo et al. [31]. This may be due to the number and type of *Metarhizium* propagules used by these authors. The number of propagules inoculated in their study was almost 17 times higher (i.e., 5 × 10^7^ conidia) than in the present study (3 × 10^6^ blastospores). Here, injection of exogenous DA, followed or not by the fungus infection, increased the number of circulating hemocytes at 24 hpi. This suggests that DA plays an important role in the cellular immune response of *R. microplus* ticks. On the other hand, in other invertebrates, treatment with DA reduced or did not affect the number of circulating hemocytes [18,37]. This contrast may be related to physiological differences between these organisms, DA dose, and time of analysis after the treatment.

Arthropods’ hemocytes are involved in phagocytosis, encapsulation, nodulation, melanization, coagulation, and production of molecules related to the immune system [15]. The phagocytic action of hemocytes against bacteria, fungi, and other foreign particles is reported to play a significant role in the immune response of ticks after infection [28,38,39,40,41,42,43]. The inhibition of DA synthesis in hemocytes of insects impaired their phagocytosis and the incubation of exogenous DA with hemocytes increased their phagocytosis [19]. These authors supported the hypothesis that DA is responsible for mediating phagocytosis by insect hemocytes. In the present study, although exogenous DA is suggested to contribute to the survival of ticks and increase the hemolymphatic number of hemocytes, after 2 h of incubation, the phagocytic index of ticks’ hemocytes was not stimulated by the previous injection of exogenous DA (Figure 1C). The present study did not investigate if longer incubation times could change *R. microplus* hemocytes’ phagocytic index. Additional evaluation of the humoral immune response and analysis of other cellular responses, such as nodulation, can help unveil unclear points about the role of DA in the immune response of ticks.

As far as we know, our study is the first report of PO activity in the hemolymph of *R. microplus*. Here, unexpectedly, the PO activity of females inoculated with *M. anisopliae* was statistically similar to untreated ticks at 72 hpi (Figure 2), suggesting that the entomopathogenic fungus did not trigger an immune response related to PO activity that would have increased their basal levels. On the other hand, the injection of exogenous DA (alone or followed by *M. anisopliae* inoculation) increased the hemolymphatic PO activity in *R. microplus* and the number of hemocytes. PO is a protein present in the cell-free hemolymph of invertebrates and according to the literature, it is activated through specific proteolytic cleavage in response to wounding or pathogenic infections [44]. PO activity has been studied in insects, such as *Rhodnius prolixus* (Insecta: Reduviidae) [45,46], *Spodoptera litura* (Insecta: Noctuidae) [23], and *Mythimna separata* (Insecta: Noctuidae) [18]. In arachnids, its activity was detected in *Polybetes pythagoricus* (Araneae: Sparassidae) [47], *O. moubata* [27], and *R. sanguineus* [28]. Feitosa et al. [28] reported increased PO activity in the hemolymph of *R. microplus* infected with the protozoan *Leishmania infantum* five days after inoculation but not at 24 or 48 hpi. The increase of PO activity has been linked to the lysis of hemocytes as they are the source of prophenoloxidase [28,44]. Despite this, the results of the present study suggested that the increase of PO activity in *R. microplus* inoculated with DA is not necessarily linked to hemocytes’ disruption.

In the present study, fluorescence microscopy analysis detected the presence of dopamine in the hemocytes of *R. microplus* ticks even under physiological conditions, suggesting that these cells can naturally produce DA. As far as we know, this is the first report of DA detection in tick hemocytes. This result supports the findings of [19], who also detected DA in hemocytes of *C. suppressalis*. Their study reported the ability of insect hemocytes to constitutively express a DA receptor in their plasma membrane, indicating that individual hemocytes are equipped to respond to DA released by themselves or by neighboring hemocytes. Our study demonstrated for the first time the modulation of exogenous DA on the cellular immune response and survival of *R. microplus* inoculated with *M. anisopliae*. These findings have a significant positive impact on the knowledge of the immune response of ticks challenged with entomopathogenic fungi.

## Figures and Tables

**Figure 1 jof-07-00950-f001:**
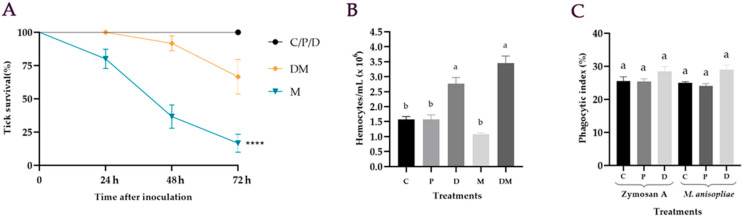
Influence of dopamine on the survival and cellular immune response of *Rhipicephalus microplus* inoculated with *Metarhizium anisopliae*. (**A**) Effect of dopamine on the survival of *R. microplus* inoculated with *M. anisopliae*. Average survival percentage and standard error of *R. microplus* females inoculated with *M. anisopliae* blastospores according to Log-rank (*p* < 0.0001). Representative experiments of two independent repetitions are shown, where (****) represents a statistical difference between M and DM (*p* < 0.0001). (**B**) *R. microplus* hemocytes quantification. Data are presented as the average number and standard error of *R. microplus* hemocytes per mL of hemolymph 24 h after injection of dopamine (1025 ng μL^−1^) and inoculation of *M. anisopliae* (10^6^ blastospores mL^−1^). Bars with the same letter do not differ statistically according to the one-way ANOVA followed by the Tukey’s test (*p* > 0.05). C: untreated ticks; P: ticks injected with phosphate buffer solution; D: ticks injected with dopamine; M: ticks inoculated with *M. anisopliae*; DM: ticks inoculated with dopamine and the fungus. (**C**) Phagocytic index of *R. microplus* hemocytes two hours after incubation with *M. anisopliae* blastospores or *Saccharomyces cerevisiae* (Zymosan A; Sigma-Aldrich, St. Louis, MO, USA). Hemocytes were collected from C: untreated tick females, P: ticks previously injected with phosphate buffer, or D: dopamine (1025 ng μL^−1^) 24 h after injection. Data were presented as means and standard error. The phagocytic index was determined two hours after in vitro incubation as a ratio of phagocytic and non-phagocytic hemocytes. Bars with the same letter do not differ statistically according to the Kruskal–Wallis test followed by the Dunn’s test (*p* > 0.05).

**Figure 2 jof-07-00950-f002:**
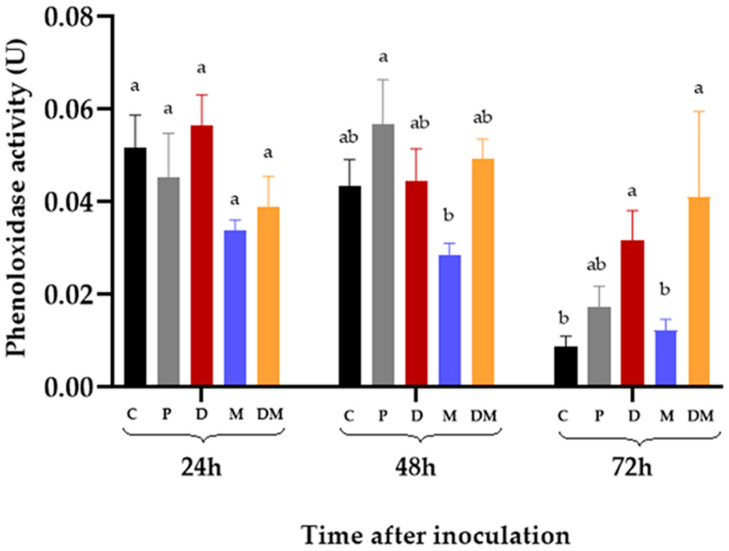
Changes in the phenoloxidase (PO) activity (U) in the hemolymph of *Rhipicephalus microplus* after injection of dopamine and inoculated *Metarhizium anisopliae*. Data were represented as means and standard error. Bars with the same letter at the same time did not differ statistically after two-way ANOVA followed by Tukey’s test (*p* > 0.05). C: untreated ticks; P: ticks injected with phosphate buffer solution; D: ticks injected inoculated with dopamine (1025 ng μL^−1^); M: ticks inoculated with *M. anisopliae* (10^6^ blastospores mL^−1^); DM: ticks injected with dopamine and *M. anisopliae*.

**Figure 3 jof-07-00950-f003:**
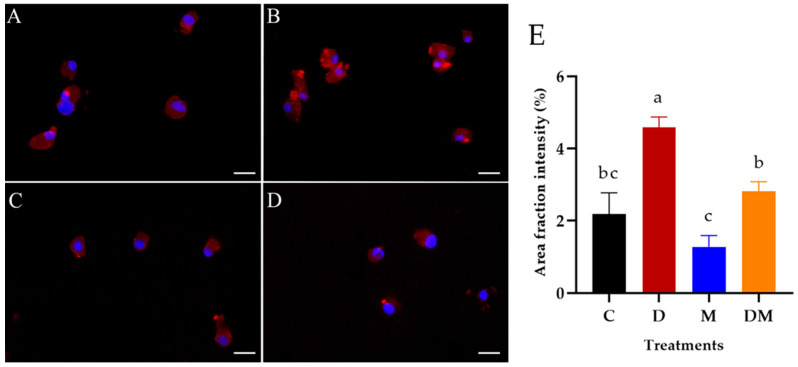
Detection of dopamine in the hemocytes of *Rhipicephalus microplus* tick females 24 h after injection of dopamine and *Metarhizium anisopliae*. (**A**) Untreated ticks (Control); (**B**) Ticks injected exclusively with dopamine; (**C**) Ticks inoculated exclusively with *M. anisopliae* blastospores; (**D**) Ticks injected with dopamine and *M. anisopliae* blastospores. Dopamine was visualized by fluorescence microscope (BX 51; Olympus, Tokyo, Japan) in the hemocytes (Alexa Fluor 594; red). Nuclei were counterstained with DAPI (blue). Scale bars = 10 μm. (**E**) Average fluorescence intensity (marked area) percentage and standard error of dopamine in *R. microplus* hemocytes calculated using the ImageJ software (National Institute of Health, Bethesda, MD, USA). Bars with the same letter do not differ statistically according to the one-way ANOVA followed by the Tukey’s test (*p* > 0.05).

## Data Availability

The data presented in this study are available on request from the corresponding author.

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
