# Peer review of "How Dopamine Influences Survival and Cellular Immune Response of *Rhipicephalus microplus* Inoculated with *Metarhizium anisopliae"

_jof, 2021, doi:10.3390/jof7110950_

Round 1

Reviewer 1 Report

Corrêa et al. is sought to evaluate the effects of dopamine on the tick immune response and survival. The results indicated that exogenous dopamine increased the host cellular and humoral immune responses, which finally increased the tick survival after infection caused by entomopathogenic fungus. Overall, this work is interesting and expands our understanding of mechanism behind the fungus-host interaction.

My concerns:

  1. The figures in manuscript should be re-organized or combined to make them concise, particularly Fig. 1, 2 and 3. It is suggested that the left panel in Fig. 4 should be changed into bar graph.
  2. Line 362-364. These sentences implied that exogenous dopamine has potential application in the tick control. However, as author’s statements, exogenous dopamine increased the host immune responses. There seems a little conflict between two statements. More attention and discussion must be added to make this suggestion reasonable.
  3. In Fig.5. The images in bright field should included.

     4.Line 122. Additional ‘.’ Should be deleted

Author Response

Response to Reviewer 1 Comments

Manuscript ID: jof-1450554

Title: How Dopamine Influences Survival and Cellular Immune Response of Rhipicephalus microplusInoculated with Metarhizium anisopliae

General comments from the authors

We would like to thank the reviewers for their very helpful comments on our manuscript. It is our opinion that incorporating the suggestions below has considerably improved the manuscript. All changes made to the manuscript are marked in the text.

Point 1:The figures in manuscript should be re-organized or combined to make them concise, particularly Fig. 1, 2 and 3. It is suggested that the left panel in Fig. 4 should be changed into bar graph.

Authors’ response 1: Thank you for your suggestion. Formerly figures 1, 2, and 3 are now combined in figure 1 (1A, 1B, and 1C). Figure 2, formerly figure 4, was rebuilt as a bar graph.

Point 2: Line 362-364. These sentences implied that exogenous dopamine has potential application in the tick control. However, as author’s statements, exogenous dopamine increased the host immune responses. There seems a little conflict between two statements. More attention and discussion must be added to make this suggestion reasonable.

Authors’ response 2: We thank you for this comment. We believe the reviewer is referring to former lines 357-361. The statement in these lines was obviously unclear in the original version of the manuscript. We apologize. The sentence is now re-written to avoid any conflict. Please refer to page 10, lines 633 - 635.

Point 3: In Fig.5. The images in bright field should be included.

Authors’ response 3: Unfortunately, while performing the fluorescence experiments the bright-field photo was not taken. In fact, recent studies such as Conceição et al. (2021) that was published in Scientific Reports (https://doi.org/10.1038/s41598-021-98738-7) also did not include bright-field images in the fluorescence microscopy analysis (please refer to figure 5 of Conceição et al. 2021). Additionally, as the Journal of Fungi gave us five days to respond to the reviewers’ comments, there was not enough time to repeat the experiment and obtain the bright-field images. If, on the other hand, the reviewer and the editor understand that these images are crucial for the publication of our manuscript, we will be happy to repeat the experiment, but it will be necessary more time to do it.

Point 4: Line 122. Additional ‘.’ Should be deleted.

Authors’ response 4: Thanks! The additional “.” was removed. Please refer to page 3, line 129.

Reviewer 2 Report

Please see attached a file.

Author Response

Response to Reviewer 2 Comments

Manuscript ID: jof-1450554

Title: How Dopamine Influences Survival and Cellular Immune Response of Rhipicephalus microplusInoculated with Metarhizium anisopliae

General comments from the authors

We would like to thank the reviewers for their very helpful comments on our manuscript. It is our opinion that incorporating the suggestions below has considerably improved the manuscript. All changes made to the manuscript are marked in the text.

Major review

Point 1: In subsection 2.3, 3 μL aliquots of blastospore suspension, DA solution or controls were used to inoculate ticks between the scutum and capitulum. Please mention the method of inoculation, dipping or injection. Don't save the key word at the first time of its presence even if a reference was cited.

Authors’ response 1: We thank you for this comment. The method of injection was mentioned in the material and methods. Please refer to the page 3, line 119 - 120.

Point 2: In subsection 2.3, please describe how to assess phagocytic index, which is not defined in the text. By the way, is a 2 h interaction of fungal infection with added DA or controls sufficient to induce differences of phagocytic index between the treatments? Is it possible for an insufficient interaction to result in no difference in phagocytic index between the treatments? Line 136, how long were the fungal cells incubated with tick hemocytes? 2 h? IMPORTANT!

Authors’ response 2: We believe that the reviewer is referring to subsection 2.5. This item is now re-written to explain how the phagocytic index was assessed. Please refer to page 4, line 147. Regarding the time of interaction between the cells and the fungi (i.e., 2h): tests performed by our research group with Ixodes (that is also a hard tick) showed no difference in the phagocytic index assessed after 2h or 6h (FIOROTTI et al., 2022 https://doi.org/10.1016/j.dci.2021.104234), however, we understand that only performing new experiments could prove that more hours of incubation result in a different outcome. This subject was covered in the discussion section. Please refer to page 9, lines 595-598.

Point 3: Figures: add right labels to X axes, such as 'Time (h) after inoculation' for Fig. 1.

Authors’ response 3: Labels to X axes were added to all figures.

Point 4: Figure 4: A three-group bar chart taking place of Fig. 1A would be better to show statistical differences over time after different treatments. In this case, Fig. 4B is unnecessary.

Authors’ response 4: Figure 2, formerly figure 4, is now rebuilt as a bar chart. Figure 4B was excluded.

Point 5: Figure 5: A bar chart showing red signal intensity values of DA, which can be measured using online ImageJ software, would help to understand the differences between treatments.

Authors’ response 5: A bar graph has been added to this figure to show DA red signal intensity values. The quantification was made using the ImageJ software, as per your suggestion. Please refer to page 5, lines 201-212 and Figure 3.

Point 6: Exogenous DA was shown affecting tick survival and increase the number of hemocytes in tick hemolymph. This is seemingly conflicting with subsequent observation on phagocytic index of tick hemocytes not enhanced by exogenous DA applied before co-incubation with fungal cells. An expanded discussion on the conflict should be informative.

Authors’ response 6: The authors agree that the reviewer’s statement is important and decided to include a sentence covering this issue. Please refer to page 9, lines 595-598.

Minor review

Point 1: Change 'effect of ... in ...' to 'effect of ... on ...' throughout the text.

Authors’ response 1: The manuscript text has been checked and it is now re-written.

Point 2: It is not appropriate to say INOCULATION with DA or buffer (control). That term suit to applied inocula or propagules, such as fungal cells. The better word is inject or injection. If host inoculation relies upon injection, hour post-injection (can be defined as hpi) is better than hour after inoculation throughout the text. The defined hpi would favor the writing as concise as possible.

Authors’ response 2: We appreciate your suggestion. The manuscript text is now re-written with the more appropriate terms. Please refer to page 1, lines 25 e 26; page 3, lines 116, 117, 118, 119, 122, 127 and 128; page 4, lines 15, 152, 154, 172 and 184; page 5, lines 230, 231, 238, 239 and 241; page 6, lines 380, 382, 383, 387, 388, 395, 397 and 398; page 7, lines 436, 439, 442, 449, 452 and 454; page 8, lines 502, 509 and 510; page 9, lines 562, 577, 578 and 594; page 10, lines 607, 609 and 620.

Point 3: Repeated proof readings are needed to improve grammar minimize.

Authors’ response 3: The grammar in the manuscript text has been checked.
